

# Gradual pollen presentation in *Vaccinium corymbosum* 'Bluecrop': an adaptive mechanism to improve pollination efficiency and outcrossing

Chunzhi Zhou[1], Yalong Yu[1], Yuwei Liu[1], Shanlin Yang[1] and Yanfeng Chen[2]

[1] School of Landscape Architecture, Changchun University, Changchun, Jilin, China
[2] School of Geography and Tourism, Qufu Normal University, Rizhao, Shandong, China

## ABSTRACT

Gradual pollen presentation is a plant reproductive mechanism to improve pollination efficiency and accuracy and promote outcrossing. *Vaccinium corymbosum* 'Bluecrop' has a typical gradual pollen presentation mechanism. 'Bluecrop' exhibits an inverted bell-shaped flower with a white coloration. By investigating the flower syndrome, pollination characteristics, pollination efficiency, and breeding system of 'Bluecrop', this study aims to explore the adaptive significance of these traits. The results showed 'Bluecrop' released pollen gradually through anther poricidal dehiscence. Among different pollinators, *Apis mellifera* and *Bombus* can pollinate effectively, and the mechanism of gradual pollen presentation significantly improved the efficiency of pollen transfer. This characteristic limits the amount of pollen removed by the pollinators and prolongs pollen presentation, thus attracting more pollinators and thereby increasing male fitness. The nectar secretion of 'Bluecrop' is gradual, with a large nectar production and a long phase of nectar secretion, enhance visitation frequencies and the chances of successful pollination. At the same time, campanulate corolla can protect pollen as well as nectar from waste due to environmental factors and other effects. The breeding system of 'Bluecrop' relies mainly on outcrossing because of its low affinity for self-fertilization and good interaction with pollinating insects. Thus, the special floral syndrome and the mechanism of secondary pollen presentation are significant in improving pollination efficiency and promoting the reproductive success of 'Bluecrop' by outcrossing. It can provide a certain theoretical basis for the future propagation breeding of 'Bluecrop'.

## INTRODUCTION

Pollen dispersal in most flowering plants is accomplished with the help of pollinators. Pollinators are the most numerous and useful pollinators group among all pollinators and play an important role in pollination and reproduction of flowering plants (*Kearns, 2004*). Bees are essential for pollen transfer and fruit production in many crops and their visit

Corresponding author
Shanlin Yang, yangsl@ccu.edu.cn

patterns can be influenced by floral morphology (*Courcelles, Button & Elle, 2013*). The interaction between plant floral traits and pollinators is considered an important driving force for plant evolution (*Pauw, Stofberg & Waterman, 2009*;*Vallejo-Marín, 2019*; *Zhou et al., 2023*). The continuous secretion of nectar is the continuous attraction of flowers to pollinators (*Chabert et al., 2018*). Nectar is a reward that flowering plants provide to their pollinators (*Luo, Zhang & Renner, 2008*). Therefore, plants and pollinators form a mutually adapted synergistic relationship, and the activity patterns of pollinators are closely related to the flowering phase, flower opening dynamics, and nectar secretion dynamics of plants (*Fenster et al., 2004*).

The male organ of angiosperms, the morphological structure of anthers and their mode of dehiscence have special traits, tomato anthers dehiscence is medial longitudinal, whereas tobacco is lateral longitudinal (*Du, 1987*). Angiosperms are constantly evolving flower forms. The reproductive organs and mating biology of angiosperms exhibit greater variety than those of any other group of organisms (*Harder & Routley, 2006*). The flower morphology of angiosperms is constantly changing (*Cardinal, Buchmann & Russell, 2018*). As an important part of the stamen, it contains reproductive and nutritional tissues related to the formation and release of pollen tetrads. In many bee-pollinated flowers, bees produce vibrations that travel through flower tissues (mainly anthers containing pollen), causing pollen to be ejected from small openings (holes or gaps) in the tips of the stamens (*Brito et al., 2020*; *Pritchard & Vallejo-Marín, 2020*; *Kemp & Vallejo-Marín, 2021*). Pollen is released at the right time after maturation to complete pollination through selfing or outcrossing, thus ensuring a smooth pollination and fertilization process. Anther dehiscence, as an important feature of late flower development, if affected by external environmental factors and other influences, such as climate leading to imperfect or complete anther dehiscence, the pollination outcome will be seriously affected (*Ding et al., 2013*). As the final stage of stamen development, whether anther dehiscence is completed on time affects whether pollen can reach the stigma in time, a key factor affecting reproductive success (*Wang et al., 2008*; *Huang et al., 2014*).Therefore, anthers play an important role as special floral trait, and the gradual pollen presentation mechanism of anthers improves pollination efficiency and ensures plant reproduction.

The pollen dispensing mechanism can control the number of divisions that pollinators take away from a packing unit in a single visit through some specific floral morphology and structure (*Li, 2013*). Anther characteristics associated with pollen distribution mechanisms, such as anther apertures (*Du, 1987*; *Falcão, Schlindwein & Stehmann, 2016*; *Vallejo-Marín, 2019*), secondary pollen presentation (*Wang, 2010*; *Yang et al., 2019*; *Xu et al., 2021*), and anther appendages (*Han et al., 2008*), have been well studied by many authors. Many plant species have evolved floral trait that restrict pollen access (*De Luca & Vallejo-Marín, 2013*; *Ashman et al., 2004*). It is known that the pollination capacity of most plants in nature is closely related to the amount of pollen in a flower as well as the mode of anther dehiscence, the level of pollination ability affects the results, and pollination is a necessary process for plants to produce fruits (*Song et al., 2013*). Because pollen is released through different modes of anther dehiscence, the efficiency of pollen dispersal can vary greatly, different ways of anther dehiscence result in different rates of pollen propagation

(*Bernhardt, William & Richard, 1996*). The pollen dispensing mechanism can well explain the relationship between plants and pollinators, the extent to which plants should restrict their rate of pollen presentation will depend on pollinator visit rates—restricting pollen presentation when pollinator visits are rare would result in lost mating opportunities and wasted pollen production (*Xiao, 2015*; *Minnaar et al., 2019*). The pollen dispensing mechanism is a special configuration of the pollen presentation time, because by adjusting the pollen presentation time, pollen can be distributed to different pollinators, thus reducing the unreliability of pollen transfer and increasing the chance of successful pollen deposition on the stigma. Thus, the analysis of relevant floral structures, combined with factors such as the mating system of plants, will help to accurately reveal the adaptive significance of the pollen dispensing mechanism, and continue to refine the shortcomings of pollen presentation theory.

*Vaccinium corymbosum* 'Bluecrop' in the family Ericaceae. There are fewer reports on its pollination mechanism, and there are still some limitations in pollen presentation theory until now. 'Bluecrop' has a special pollen presentation mechanism, nectar secretion mode, and petal unfolding mode. Therefore, we investigated the floral syndrome, pollination characteristics, flower-visiting insect, and foraging behavior of flower-visiting insects to explore the influence of its special floral traits on the pollination mechanism, go ahead and keep refining the shortcomings of pollen presentation theory. The purpose of this study: (1) what pollen distribution mechanisms do 'Bluecrop' have during flowering; (2) different pollinating insects have different pollination adaptations to 'Bluecrop'; (3) how nectar presentation strategies and pollen dispersal patterns affect the frequency and behavior of pollinators visiting flowers. Therefore, we focus on the special floral traits of 'Bluecrop', and this study will help to understand the interaction between floral traits and pollination adaptations.

## MATERIALS AND METHODS

### Study site and species

The experimental site was located in a blueberry nursery within Changchun City (125°18′ E, 43°49′ N), Jilin Province (only blueberry populations were present in the nursery, and no other plants interfered with the experimental populations). It has an average annual temperature of 4.6 °C, average annual precipitation of 600–700 mm. During the flowering season of 'Bluecrop', rainfall is 150 mm from May to June. 100 plants of 5-year-old 'Bluecrop' were selected. 'Bluecrop' is a cultivar of *Vaccinium corymbosum* Ericaceae, also know as "northern highbush blueberry". The adult height of 'Bluecrop' was 1.2 ± 0.3 m, with a canopy size (east-west) of 1 ± 0.2 m and a canopy size (south-north) of 1 ± 0.2 m; there were 8 ± 2 flowers per inflorescence.

### Categorization of flower life into four stages

According to the flowering dynamics of 'Bluecrop', the flowering process could be divided into four phase, namely, Phase I, just before flower anthesis; Phase II, flower just opened with the aperture not completely opened; Phase III, flower with the aperture completely opened; and Phase IV, the petals are all falling off (Fig. 1).

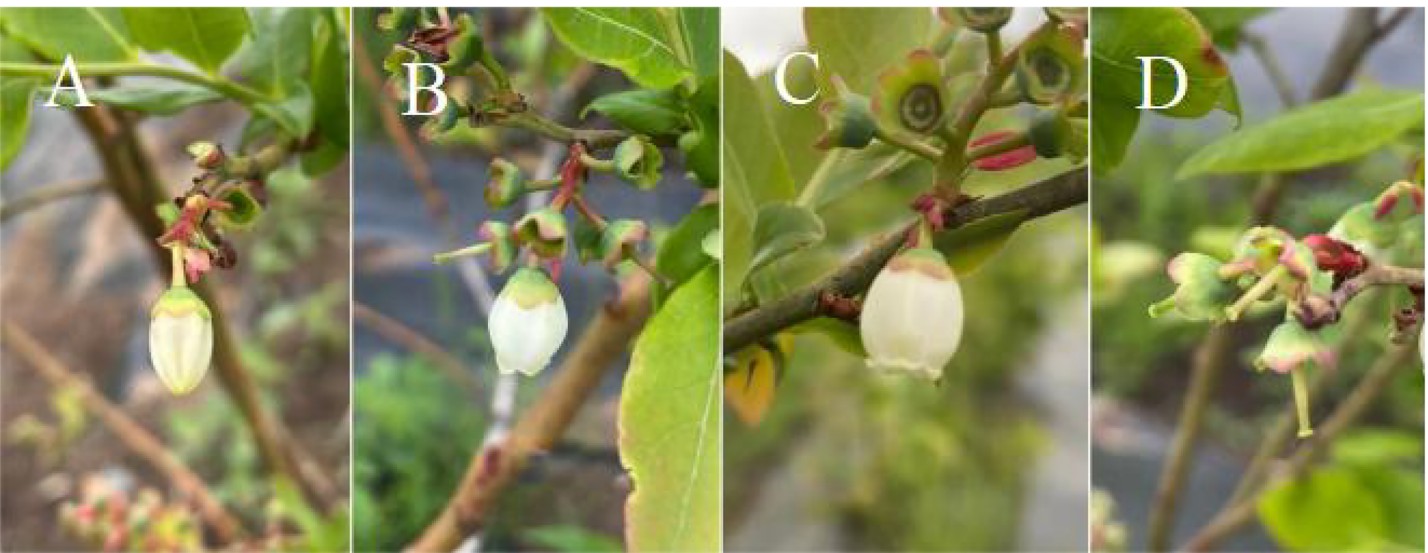

**Figure 1** **Single flower blooming dynamics of 'Bluecrop'.** (A) Phase I (B) Phase II (C) Phase III (D) Phase IV.

## Dynamics of pollen release in relation to the flower age

Thirty unopened flowers were randomly selected from the 'Bluecrop' population
for bagging, and the status of each flower was tracked and observed every hour. The
process of anther dehiscence was observed with a body vision microscope, and the process
and manner of anther dehiscence were recorded, with a focus on whether there were
changes in pistils and stamens and whether the anthers were dehiscent to release pollen.
Pollen collection requires a soft brush to shake pollen off and a pollen tube to collect pollen
tetrads. 1 mL of 1% sodium hexametaphosphate solution was added dropwise to the
pollen, and the volume was fixed to 2 mL. The pollen was covered and shaken to keep the
pollen in suspension. A drop of the suspension was aspirated on a hemocyte counting plate
using a micro-sampler with a volume of 20 μL. The number of pollen tetrads was observed
and counted under microscope. The process was repeated six times and the average was
taken.

## Observation of pollen tetrad and stigma morphology

Ten flowers of essentially uniform development were randomly selected from the
'Bluecrop' population, and after the flowers opened, the bags were removed, and the 10
flowers were fixed in 50% FAA fixative, respectively. Gradient concentrations of ethanol
(65%–75%–85%–90%–100%–100%–100%) dehydration were applied before electron
microscopy scanning, each time for 15 min. All pollen tetrads and stigmas were dried at
the critical point. The dried pollen tetrad samples were fixed on the sample tray with
conductive double-sided tape, and the stigma was coated with gold palladium. The pollen
tetrad and stigma morphology were observed with a JSM-6510 (Japan) scanning electron
microscope (Changchun University).

## Pollen viability and stigma receptivity in relation to flower age

During the flowering phase of the population, flowers of 'Bluecrop' were collected from four different phase. Pollen from ten single flowers from each phase was selected for testing for the assay. Their anthers were taken and placed on slides with 1–2 drops of 0.5% TTC solution, placed in an incubator at 35 °C for 15 min, and the degree of staining was observed under the microscope. The pollen-stained red was considered to have the strongest viability, and the pollen-stained light red was considered to have weaker vitality, and pollen that was not stained was considered to non-viable or sterile.

During the flowering phase of the population, flowers of 'Bluecrop' were collected from four different phases, ten stigmas from each phase were selected for testing, and the stigma was removed intact, placed on a concave slide, and dripped into the reaction solution of benzidine-hydrogen peroxide (1% benzidine: 3% hydrogen peroxide: water) = 4:11:22). If the column head appears blue with a large number of bubbles around, it was considered to be highly receptive (denoted as +++); if the column head appears light blue, surrounded by more air bubble, it was considered to be relatively weak receptivity (denoted as ++); if the column head appears light blue, with a small number of bubbles around, it was considered to be relatively weak receptivity (denoted as +); if the column head has no color change and no bubbles around, it was considered to be not receptivity (denoted as +/−) (*Zhang et al., 2022*; *Dafni, 1992*; *Dafni, Kevan & Husband, 2005*; *Baptiste & Fang, 2023*).

## Foraging behavior and pollination behavior of insect foragers

We chose sunny weather for observations to avoid the effects of rain on the flower-visiting behavior of insects from June 6 to June 8, 2021. Ten inflorescences of 'Bluecrop' that had opened were randomly selected within the population and marked to observe the external morphology of pollinators, flower-visiting time, number of flower-visiting times, and their flower-visiting behavior. The pollen carried by the insects was placed on slides and observed by light microscope, and the insects that finished pollination were defined as effective pollinators. Species identification by external morphology of pollinators.

Calculation of pollination efficiency of pollinators. For observation, two flowers with essentially synchronous development and unopened flowers were randomly tagged and bagged on each of 10 plants (20 flowers in total). They were divided into two groups for anther pollen dispersal. The anther of one group were removed directly, and the other group was removed after a visit by pollinators. The total number of pollen on the anthers of the two groups of flowers were counted under a light microscope. The amount of pollen removed by pollinators after a flower visit is the difference between the two groups of pollen counts.

In order to detect the number of pollen deposited on the stigma after a visit by an insect, 30 flowers with essentially synchronized development and unopened flowers were randomly marked on 15 plants. The stigma was crushed after a visit by an insect and stained with saffranin and fixed to 2 mL, and the pollen count was counted under a light microscope. Insect pollination efficiency is the amount of pollen deposited on the stigma by a single visit of the insect divided by the amount of pollen removed by the insect in a single visit.

### Dynamics of nectar secretion and flower visitation frequency by insect foragers in relation to flower age

In sunny weather at the observation site, a single flower that will open the following day was randomly labelled on each of 30 plants and observed continuously from 7:00 to 19:00 when pollinators appeared at the early flowering stage. The frequency of flower visits by different pollinators was recorded until the end of the flowering stage.

On 30 plants that developed almost simultaneously, each plant was randomly selected to bag an unopened flower. From the beginning to the end of the flowering period. Measurement of flowers needing to be bagged to avoid insect influence on nectar volume. Reference to Corbet's methodology (*Corbet, 2003*). Nectar volume was measured every 24 h with a 5 µL micropipette. The relationship between nectar secretion dynamics and the frequency of flower visits by pollinators was analysed on the basis of their measurements.

### Pollination experiment

Reference to Castro's methodology (*Castro, Silveira & Navarro, 2008*). The type of breeding system was detected by an artificial bagging experiment. A total of 60 plants were randomly selected with the same development, and four flowers were randomly selected on each plant for the following four treatments. (1) Open pollination: detection of pollination and fruiting under natural conditions. (2) Bagged without emasculation: tested for the presence of active self-pollination. (3) Artificial autogamy (bagged after pollination to exclude interference): testing self-pollination for affinity. (4) Artificial xenogamy (bagged after pollination to exclude interference): detection of fruiting in artificial xenogamy pollination. Three replicates were set for each treatment.

### Statistical analysis

Experimental data are represented as the mean ± SD (standard deviation), fruiting data are represented as the mean. Pollen counts, pollen viability, number of insect visits, nectar secretion, and pollination experiments were statistically analyzed using SPSS 19.0 software (IBM SPSS, Armonk, NY, USA). When the statistic was significant, one-way ANOVA was used to compare the differences based on the Duncan's multiple range test ($p < 0.05$). Analysis of pollination experiment data. A $p < 0.05$ was considered statistically significant. Making data into charts using origin 2018.

## RESULTS

### Dynamics of pollen release in relation to the flower age

The number of pollen remaining in each period is shown in the figure (Phase I, 24,100 ± 278; Phase II, 19,800 ± 237; Phase III, 10,500 ± 147; Phase IV, 2,080 ± 132) (Fig. 2). Instead of releasing all of the pollen tetrads at once through the anthers, 'Bluecrop' released a portion of the pollen tetrads at each phase, and the number of pollen in a single flower of 'Bluecrop' was 24,100 ± 278.

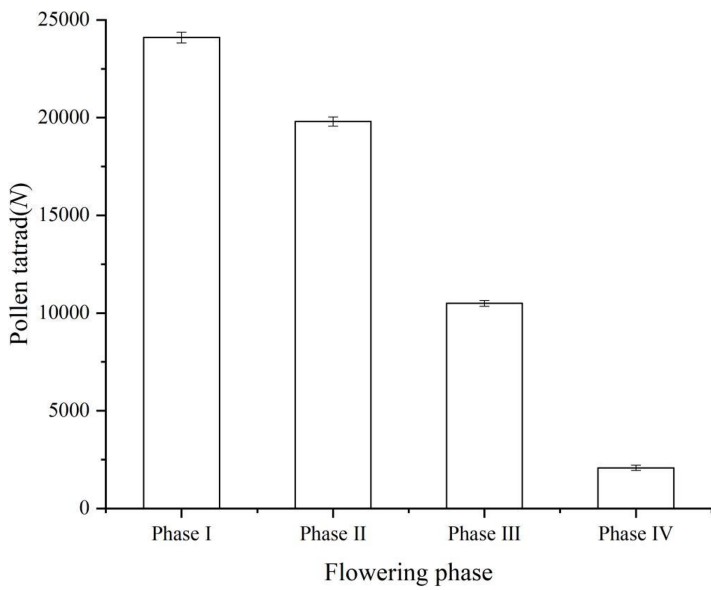

**Figure 2 Pollen tetrads dispersal process during the opening of a single flower of 'Bluecrop'.** The number of pollen remaining in each period is shown in the figure (Phase I, 24,100 ± 278; Phase II, 19,800 ± 237; Phase III, 10,500 ± 147; Phase IV, 2,080 ± 132). 

## Observation of pollen tetrad and stigma morphology

The surface of the stigma of 'Bluecrop' was smooth, and its stigma was poricidal-like in the center and radiated five fissures of different shapes (Fig. 3A); the anthers were elongated (Figs. 3B and 3C); the pollen tetrads were compound pollen with fine folds on the surface (Fig. 3D).

## Pollen viability and stigma receptivity test results in relation to flower age

Pollen viability was of 52.8% (±4.66%) at flower phase I, peaked at flower phase II at 79.2% (±2.59%), and decreased at flower phase III at 38.2% (±2.49%), the lowest pollen viability at phase IV was 9.4% (±1.67%) (Fig. 4). Stigma receptivity results showed (Table 1) that stigma receptivity was stronger in the phase I, strongest in the phase II, and weakest in the phase IV. Therefore, propagation culture of 'Bluecrop' is best done in the early flowering phase.

## Foraging behavior and pollination behavior of insect foragers

When a bumblebee visits a flower, the *bumbus* shaking the pollen down to the abdomen of the body, and then sends the pollen from the abdomen to the pollen-carrying foot to finish carrying and make it pollinate successfully. *Bombus* spent 30 ± 5 s on individual inflorescences and 10 ± 2 s on flowers, and effective flower visitors completed a maximum of 130 ± 6 visits between 13:00 and 14:00 each day (Fig. 5); When *Apis mellifera* visit flowers, they first extend their heads into the corolla and collect pollen on their forefeet to their hindfeet to complete the pollination process. The *Apis mellifera* spent 25 ± 5 s on a

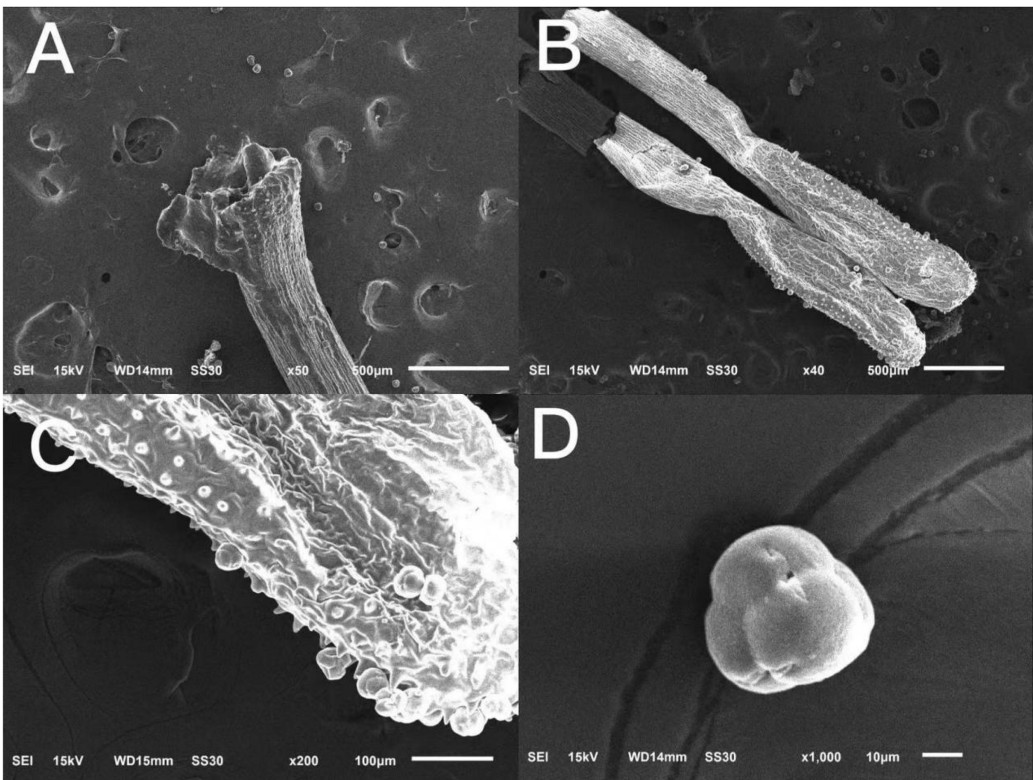

**Figure 3 Scanning picture of 'Bluecrop' flower structure with an electron microscope.** (A) Complete column head structure. (B) Complete anther structure. (C) The local surface structure of anther. (D) Morphology and structure of pollen grains.               

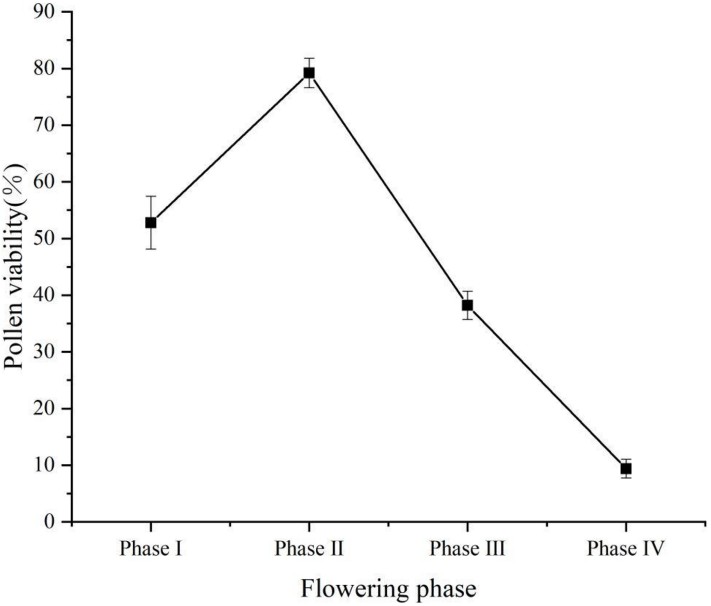

**Figure 4 Dynamic change curve of pollen vitality of 'Bluecrop' anthers.** Pollen viability was at 52.8% (±4.66%) at flower phase I, peaked at flower phase II at 79.2% (±2.59%), and decreased at flower phase III at 38.2% (±2.49%), the lowest pollen viability at phase IV was 9.4% (±1.67%).

**Table 1 'Bluecrop' stigma receptivity test.** Stigma receptivity results showed that stigma receptivity was stronger in the phase II, strongest in the phase I, and weakest in the phase IV.

| Flowering phase | Phase I | Phase II | Phase III | Phase IV |
|---|---|---|---|---|
| Stigma receptivity | ++ | +++ | + | +/− |

Note:
The stigma receptivity strength: Strongest (+++); Stronger (++); Weak (+); Weakest (+/−).

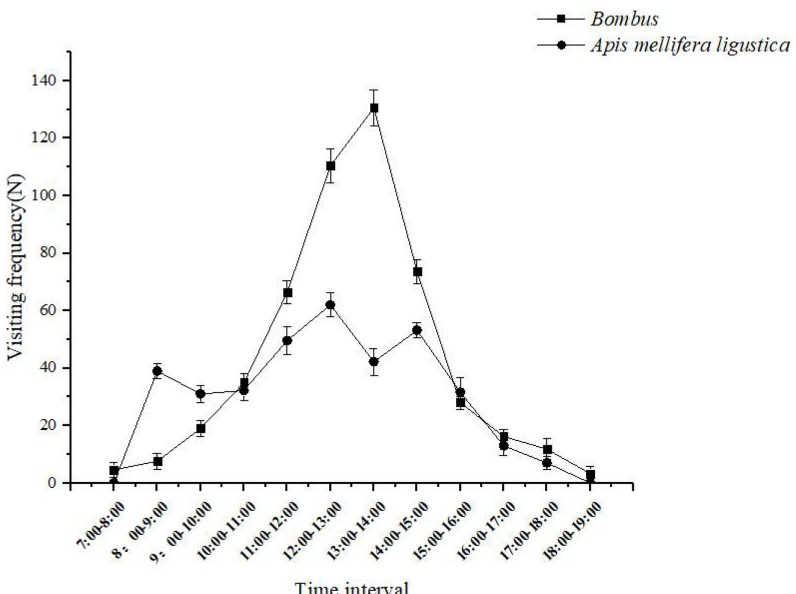

**Figure 5 The daily change in the frequency of effective flower visits of 'Bluecrop' pollinators.** *The Bombus spent 30 ± 5 on individual inflorescences and 10 ± 2 s on flowers, and effective flower visitors completed a maximum of 130 ± 6 visits between 13:00 and 14:00 each day. The Apis mellifera ligustica spent 25 ± 5 on a single inflorescence and 7 ± 3 s on a flower, and pollinators completed a maximum of 62 ± 4 visits between 12:00 and 13:00 daily.*

single inflorescence and 7 ± 3 s on a flower, and pollinators completed a maximum of 62 ± 4 visits between 12:00 and 13:00 daily (Fig. 5).

The number of pollen removed by *Bombus* and *Apis mellifera* after a single visit to the flowers was 4,670 ± 137 and 3,160 ± 128, respectively, and the number of pollen deposited on the stigma after a single visit was 413 ± 37 and 203 ± 18, respectively. Therefore, the pollination efficiency of the two pollinators was 8.84% and 6.42%, respectively. Among them, *Bombus* are the most efficient pollinators.

## Dynamics of nectar secretion and flower visitation frequency by insect foragers in relation to flower age

The mechanism of nectar secretion during flowering of 'Bluecrop' was gradual, with a gradual increase in the amount of nectar during the flower life from the start of pollen dispersal to 24 h after pollen release, and then a continuous decrease until it reached its lowest point at the end of flowering. As nectar production increased, the frequency of flower visits increased for two pollinators. As nectar secretion decreased, the frequency of flower visits by both pollinators decreased (Fig. 6).

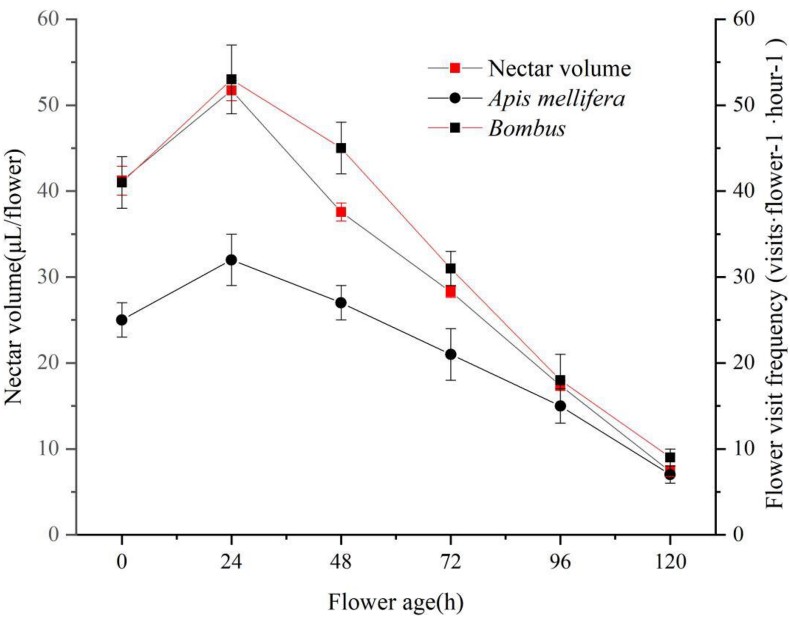

**Figure 6 'Bluecrop' nectar secretion dynamics and flower visiting frequency.** As nectar production increased, the frequency of flower visits increased for two pollinators. As nectar secretion decreased, the frequency of flower visits by both pollinators decreased.

**Table 2 Pollination experiment results of 'Bluecrop'.** 'Bluecrop' had the highest fruiting rate of xenogamy and the lowest fruiting rate of selfing.

| Treatments | Flowers (N) | Number of results (N) | Fruiting rate (%) |
| --- | --- | --- | --- |
| Open pollination | 60 | 20 | 33.3 ± 1.15[b] |
| Bagged without emasculation | 60 | 2 | 3.33 ± 0.57[c] |
| Artificial autogamy | 60 | 26 | 43.3 ± 1.53[b] |
| Artificial xenogamy | 60 | 47 | 78.3 ± 2.52[a] |

**Note:**
  Different lowercase letters indicate significant differences ($p < 0.05$).

## Pollination experiment results

'Bluecrop' had the highest fruiting rate of artificial xenogamy and the lowest fruiting rate of bagged without emasculation. There were significant differences in the fruiting rates of open pollination, artificial autogamy, and bagged without emasculation compared with artificial xenogamy ($p < 0.05$). This indicates that it has low self-fertilization affinity and essentially no autonomous self-fertilization. Fruiting is mainly dependent on pollinators as mediators (Table 2). This indicates that 'Bluecrop' is predominantly outcrossing. There is no pollen limitation for fruiting.

## DISCUSSION

### Ecological adaptation significance of integrated floral syndrome

Floral syndrome can be directly related to the pollination and evolution of plants (*Barrionuevo, Benitez-vieyra & Sazatornil, 2021*). At the same time the floral traits of plants are subject to a combination of environmental and biological constraints

(*Scheffknecht et al., 2007*; *Tang & Han, 2007*). Among them, environmental factors directly drive the adaptive evolution of floral traits (*He et al., 2005*). Plants must adapt to their environment by adjusting the structure of their flowers (*He, Wu & Jia, 2007*). 'Bluecrop' has open downward mitriform flowers, and during flowering, the plant's sexual organs keep growing inside the corolla, prolonging the time of pollen viability and high intensity of stigma fertility, and these effects are in complete agreement with the findings of *Wang & Tan (2011)* on the floral traits of *Codonopsis clematidea*. The floral traits of 'Bluecrop' can avoid pollen wastage resulting in deficiency and enable nectar secretion gradually, which is a wise decision of plant ecological adaptation in terms of resource allocation, and can also prevent nectar reduction due to rainfall, pollen being blown off by natural wind and other adverse environmental factors.

The multi-payoff strategy adopted by 'Bluecrop' on floral syndromes (pollen tetrads, flower color, nectar, anthers, *etc.*). It can be effective in increasing the frequency of pollinator visits and increasing the number and opportunity of flower visits (*Barrett, 1998*). Its flowering stage II high pollen viability and stable stigma pollinability. As well as the high coincidence of the peak period of insect flower visit with the period of highest pollen vigor and nectar secretion. It is conducive to its smooth pollination and guarantees its reproductive success (*Bingham & Orthner, 1998*; *Barrett, 2003*). Therefore, the traits of this particular flower guarantee successful pollination through pollen dispersal and nectar production. It further attracts pollinators to increase reproductive success.

## Effect of gradual pollen presentation mechanism on pollination adaptation

Nectar is a sap secreted by the nectar glands of plant flowers that attracts pollinators to take nectar and is an important factor in pollinator behaviour (*Carter & Thornburg, 2000*; *Johnson & Nicolson, 2008*). From an evolutionary perspective, plants need to allocate nectar production temporally in order to attract as many pollinators as possible for effective pollination. From an evolutionary ecological point of view, changes in the frequency of flower visits accompanying nectar dynamics are of great value in promoting allopatric pollination (*Canto et al., 2008*). In the present study, we found the presence of a gradual pollen presentation mechanism in 'Bluecrop' and also a gradual secretion of nectar. The peak period of flower visit is associated with larger nectar production and longer nectar production time, which increases the frequency of pollinator visit and increases the chance of successful pollination and pollination efficiency.

In angiosperms, the diversity of pollen's progressive presentation has attracted great attention. Gradual pollen presentation is one of the typical floral traits of plants that increase paternal fitness (*Harder & Thomson, 1989*). The pollen progressive presentation mechanism is firstly an adaptation to the number of pollinators and tends to occur in plants with abundant pollinators but low pollination efficiency (*Harder & Thomson, 1989*). In environments with a wide variety of pollinators, the pollen progressive presentation mechanism improves paternal fitness, which can reduce pollen loss under adverse environmental conditions and also reduce competition (*Liu, 2009*). Male-male competition in plants is thought to exert selection on flower morphology and on the

temporal presentation of pollen (*Castellanos et al., 2006*). Secondly, the pollen progressive presentation mechanism is also a response to pollination efficiency. The pollen progressive release would enable pollinators to take fewer pollen tetrads after one visit and avoid pollen wastage (*Thomson et al., 2000*). The anther dehiscence of 'Bluecrop' is achieved by gradually extruding pollen from the apical poricidal of the anther by contracting and squeezing the anther, which belongs to the anther poricidal dehiscence. It belongs to the "gradual pollen presentation mechanism". This mechanism limits the pollen output, so that pollinators only get a small amount of pollen in one flower visit, and more pollinators participate in the pollination process. There were also differences in the pollinnation efficiecy of effective pollinating insects in 'Bluecrop', and it is possible that this pollen progressive presentation mechanism evolved under the selection of male function.

## Interaction between breeding systems and pollinators

Pollination is an important factor affecting fruit development in highbush blueberry (*Vaccinium corymbosum* L.) (*Nagasaka et al., 2022*; *Liu et al., 2022*). The difference between the dispersal ability of pollen and the reception of the stigma, combined with the unpredictability of the pollinators's pollination behavior, can change the type of breeding system of the plant (*Xiao, 2015*). Pollinators getting a reward will cause selection pressure on the floral traits of plants, and the floral structure evolves continuously to adapt to the selection of its pollinators. It is generally believed that the floral attractants that lure insects for pollination are the color and shape of flowers, and in return pollinators will get nectar and pollen (*Murcia, 1990*). Pollinators can directly or indirectly affect plant sexual reproduction (*Campbell et al., 2010*; *Darwin, 2009*; *Ouvrard, Quinet & Jacquemart, 2017*). Accurate and efficient transfer of pollen to heterostylous stigmas not only improves male fitness, but also ensures the success of cross-pollination (*Lopes & Machado, 1999*). For plants with self-fertilization affinity, self-fertilization is produced and may produce self-fertilization decline, but, when pollen sources are lacking, self-fertilization ensures that they reproduce offspring. In contrast, 'Bluecrop' has a low affinity for self-fertilization and needs pollinators to participate in the pollination process, as well as its pollen progressive presentation mechanism for high pollen utilization and very good interactions with pollinating insects, so revealing that the breeding system of 'Bluecrop' should rely mainly on outcrossing. And the breeding culture of 'Bluecrop' is best carried out in the early flowering.

## CONCLUSIONS

Gradual pollen presentation promotes effective pollen dispersal, and for insect-pollinated plants, male fitness decreases with the amount of pollen available to the plant at one time, so most plants that rely primarily on insect pollination can improve reproductive success through gradual pollen presentation as well as gradual nectar secretion. The results of the study showed that the corolla of 'Bluecrop' monoflower faces downwards, the pollen dispersal mode is a gradual pollen presentation mechanism, the pollen grains are tetrad composite pollen, and the anther dehiscence mode is foraminal dehiscence, and the nectar

secretion mode is a gradual secretion. Floral traits of plants not only affect their attraction to pollinators and pollen walks, but are also closely linked to pollination mechanisms. Secondary pollen presentation mechanisms in angiosperms are biologically important for improving male or female fitness in plants, avoiding interference between male and female functions, and promoting cross-fertilization.

### Funding

This work was supported by the "Science Research Project of the Jilin Provincial Education Department (JJKH20230664KJ)", the "Science and Technology Development Plan Project of Science and Technology Department of Jilin Province (YDZJ202201ZYTS468)", the "Changda Researcher Climing Plan of Changchun University (ZKP202031)", and the "National Natural Sciences Foundation of China (32101262)". The funders had no role in study design, data collection and analysis, decision to publish, or preparation of the manuscript.

### Grant Disclosures

The following grant information was disclosed by the authors:
Science Research Project of the Jilin Provincial Education Department: JJKH20230664KJ.
Science and Technology Department of Jilin Province: YDZJ202201ZYTS468.
Changda Researcher Climing Plan of Changchun University: ZKP202031.
National Natural Sciences Foundation of China: 32101262.

### Competing Interests

The authors declare that they have no competing interests.

### Author Contributions

- Chunzhi Zhou conceived and designed the experiments, performed the experiments, analyzed the data, prepared figures and/or tables, and approved the final draft.
- Yalong Yu conceived and designed the experiments, performed the experiments, analyzed the data, prepared figures and/or tables, and approved the final draft.
- Yuwei Liu performed the experiments, authored or reviewed drafts of the article, and approved the final draft.
- Shanlin Yang conceived and designed the experiments, prepared figures and/or tables, authored or reviewed drafts of the article, and approved the final draft.
- Yanfeng Chen analyzed the data, authored or reviewed drafts of the article, and approved the final draft.

### Data Availability

The raw data is available in the Supplemental File.

## Supplemental Information

Supplemental information for this article can be found online at http://dx.doi.org/10.7717/peerj.17273#supplemental-information.

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
