# Peer review of "Gradual pollen presentation in Vaccinium corymbosum ‘Bluecrop’: an adaptive mechanism to improve pollination efficiency and outcrossing"

_PeerJ, doi:10.7717/peerj.17273_

## Round 0.1 · original submission · Major Revisions

The authors are requested to address the comments of all reviewers.

·

Basic reporting

The MS is about the flower and pollinaiton biology of Vaccinium. However, the authors did not efficiently mention the important differences of blueberry crop in the introduction of the MS. They did not write about tetrat pollens, the needle like anther shape etc. at the same time in line 62 they should remark that they are writing about Vaccinium and enhance this part.
The authors did not used important literatures about Vaccinium and this evaluated as a very important deficiency.
There are defective sentences in line 43, 146, 150, 158-159, 160,304 etc..
Terms leasable in line 150, jerkogamy in 212, gold sprayed (coated) in 135, pollinaitors rich environment in 302.
At the same time title did not match wtih the MS and could be used with a more impressive manner.

Experimental design

The experiment is well desiged and detailed as replicable.. However the authors did not effectively use this valuable data in results and discussion section.

Validity of the findings

The authors did not make an efficient results section. There are valuable data however the authors could not effectively use this data in results and discussion. For example, the authors did not mention SEM data in the MS and they did not make any reference to some figures and tables.
In Vaccinium there are a lot of interessting structures and authors did not mention some of these stages. For instance, the authors did not mention the tetrat stage of the pollens, the pollen dispersal stages, the anther structure etc..

The authors did not discuss the flower structure with other Vaccinium manuscripts. There are a lot of manuscripts about Vaccinium flower biology and pollination biology of Vaccinium.
Firstly, the authors should read the chapter about Blueberry in "Insect pollination of cultivated crop plants" by McGregor.
The authors have also been readed the articles listed below before preparing this MS. Because the authors did not read these articles they could not make any discussion with blueberry crop and the authors did not used the proper terms about blueberry flower biology.
-Bee visit rates vary with floral morphology among highbush blueberry cultivars (Vaccinium corymbosum L.)
-Insights into the Physiological and Molecular Mechanisms Underlying Highbush Blueberry Fruit Growth Affected by the Pollen Source
-The Xenia Effect Promotes Fruit Quality and Assists in Optimizing Cross Combinations in ‘O’Neal’ and
‘Emerald’ Blueberry.
-Insights into the Physiological and Molecular Mechanisms Underlying Highbush Blueberry Fruit Growth Affected by the Pollen Source

Because of the results section is not well studied, the discussion and conclusion sections were become deficient.

Additional comments

The authors should make an English grammar control. They used lots of very long and inverted sentences. Authors should use more clear sentences.
They should make attribution to all figures and tables for a clear MS.

Reviewer 2 ·

Basic reporting

In lines 108-120 where the site is described their is not description of where in the field the subjects are located or proximity to field edges. Hedge row vegetation can have a significant impact on pollination and pollinator populations. Especially considering there were notations on pollination behaviors starting at line 153.

Also, in addition to the annual average precipitation I would be interested in the precipitation during the floral period. Especially considering the honey bee activity impacts of precipitation. In US growing zones most of the annual precipitation comes in the same time of year as floral induction, making it a much more relevant issue for pollination studies.

Experimental design

No comment

Validity of the findings

no comment

Additional comments

Interesting premise and relevant for breeding as bluecrop is aging out of most production regions this work helps to inform the progeny based on pollination (one of the most critical production characteristics).

Reviewer 3 ·

Basic reporting

Review for PeerJ – manuscript #92427

In this manuscript, the authors led multiple experiments to describe some flower traits of one cultivar of northern highbush blueberry and some traits of foraging behavior of honey bees and bumble bees related to pollination: the dynamics of pollen release and nectar secretion in relation to the flower age, the pollen viability and stigma receptivity in relation to the flower age, the flower visitation rate by honey bees and bumble bees in relation to the hour of the day, the time spent per bee per flower visit, the number of pollen tetrads removed by bees from anthers after one visit, the number of pollen tetrads deposited by bees on stigmas after one visit, the quantity pollen limitation of plants of the experimental field monitored.

General comments
Introduction
The authors investigated many different traits in this paper, and they can mix them I a single paper, but for that, they really need to find a way to introduce all these concepts in a single story in the introduction. It is not the case at all in the present version of the manuscript. In this version, the authors focused on pollen presentation / pollen release, and most of the introduction includes vague sentences with a lot of redundancy, e.g., lines 47-52: “The interaction between plant floral characteristics… inducing effect on some insects.” The authors should also diversify their literature, they based a large part of their literature on local journals of a specific country, while the topics that they investigate are covered by many international journals, that include papers with a wider impact than the papers that they quote.

For instance, to help them for the introduction, the authors can use the paper of Ouvrard et al. (2017) (all the references are given at the end of this review): in this paper, Ouvrard et al. also covered some of the different topics investigated by the authors of the manuscript, but with oilseed rape. To introduce the concept of pollen presentation / pollen release, the authors should use reviews on this topic, rather than just a random sample of some papers working on it. They can also use Chabert et al. (2018) about the period of stigma/pistil receptivity and nectar secretion. For the concept of pollen presentation, pollen dispersal, plant male and female fitness, I recommend to the authors using and quoting the following reviews: Thomson et al. (2000), Castellanos et al. (2006), Harder and Routley (2006), Minnaar et al. (2019). As the Vaccinium genus bears flowers with poricidal anthers and requires sonication / vibrated by insects to release pollen, the authors should really introduce this concept in their introduction. For that, the authors can use the following reviews: Luo et al. (2008), De Luca and Vallejo-Marín (2013), Falcão et al. (2016), Cardinal et al. (2018), Vallejo-Marín (2019, 2022), Brito et al. (2020), Pritchard and Vallejo-Marín (2020), Kemp and Vallejo‐Marín (2021), Vallejo-Marin and Russell (2023). But I do not think that the reference to the pollination syndrome is relevant any more to report in the introduction, see Ollerton et al. (2009) and Dellinger et al. (2019).

In the last paragraph of the introduction, the authors should present their aims rather than ‘hypotheses’. This manuscript is more descriptive than hypothesis-driven So they should present what they describe, their aims of description (synthetized at the beginning of the review). In addition, the cultivar Bluecrop should not be introduced in the introduction, but only in the materials and methods. In the introduction, the authors should only refer to Vaccinium corymbosum.

=> I recommend a complete rewriting of the introduction, by following the previous suggestions.


Semantics
Please, use ‘flower trait’ instead of ‘flower characteristic’.
Please, refer to “pollen tetrads” for Vaccinium and not “pollen grains”.
Please, use the word ‘receptive’ instead of ‘licensable’ about stigma receptivity.


Some minor comments
Line 109, field or nursery instead of ‘plantation’

Line 110, “with an average annual temperature… a freezing phase of 150 days”: not especially useful for the study, it can be removed.

Line 112, ‘canopy size’ instead of ‘crown width’, but the sentence “The adult height…(south-north) of 1 ± 0.2 m” can even be removed as this information is useless for the study.

Line 114, replace “the inflorescence was a raceme with” with “there were”. Remove the end of the sentence “the flower was white…dark blue”, also useless.


References
Ashman, T. L., Knight, T. M., Steets, J. A., Amarasekare, P., Burd, M., Campbell, D. R., et al. (2004). Pollen limitation of plant reproduction: ecological and evolutionary causes and consequences. Ecology, 85(9), 2408-2421.

Brewbaker, J. L. (1967). The distribution and phylogenetic significance of binucleate and trinucleate pollen grains in the angiosperms. American Journal of Botany, 54(9), 1069-1083.

Brito, V. L. G., Nunes, C. E. P., Resende, C. R., Montealegre-Zapata, F., & Vallejo-Marín, M. (2020). Biomechanical properties of a buzz-pollinated flower. Royal Society Open Science, 7(9), 201010.

Cardinal, S., Buchmann, S. L., & Russell, A. L. (2018). The evolution of floral sonication, a pollen foraging behavior used by bees (Anthophila). Evolution, 72(3), 590-600.

Castellanos, M. C., Wilson, P., Keller, S. J., Wolfe, A. D., & Thomson, J. D. (2006). Anther evolution: pollen presentation strategies when pollinators differ. The American Naturalist, 167(2), 288-296.

Chabert, S., Lemoine, T., Cagnato, M. R., Morison, N., & Vaissière, B. E. (2018). Flower age expressed in thermal time: is nectar secretion synchronous with pistil receptivity in oilseed rape (Brassica napus L.)?. Environmental and Experimental Botany, 155, 628-640.

Corbet, S. A. (2003). Nectar sugar content: estimating standing crop and secretion rate in the field. Apidologie, 34(1), 1-10.

Courcelles, D. M. M., Button, L., & Elle, E. (2013). Bee visit rates vary with floral morphology among highbush blueberry cultivars (V accinium corymbosum L.). Journal of Applied Entomology, 137(9), 693-701.

Dafni, A. (1992). Pollination Ecology: A Practical Approach. Oxford University Press.

Dafni, A., Kevan, P. G., & Husband, B. C. (2005). Practical Pollination Biology. Enviroquest.

Dellinger, A. S., Chartier, M., Fernández‐Fernández, D., Penneys, D. S., Alvear, M., Almeda, F., et al. (2019). Beyond buzz‐pollination-departures from an adaptive plateau lead to new pollination syndromes. New Phytologist, 221(2), 1136-1149.

De Luca, P. A., & Vallejo-Marín, M. (2013). What's the ‘buzz’about? The ecology and evolutionary significance of buzz-pollination. Current Opinion in Plant Biology, 16(4), 429-435.

DeVetter, L. W., Chabert, S., Milbrath, M. O., Mallinger, R. E., Walters, J., Isaacs, R., et al. (2022). Toward evidence-based decision support systems to optimize pollination and yields in highbush blueberry. Frontiers in Sustainable Food Systems, 6, 1006201.

Falcão, B. F., Schlindwein, C., & Stehmann, J. R. (2016). Pollen release mechanisms and androecium structure in Solanum (Solanaceae): Does anther morphology predict pollination strategy?. Flora, 224, 211-217.

Harder, L. D., & Aizen, M. A. (2010). Floral adaptation and diversification under pollen limitation. Philosophical Transactions of the Royal Society B: Biological Sciences, 365(1539), 529-543.

Harder, L.D., & Routley, M.B. (2006). Pollen and ovule fates and reproductive performance by flowering plants. In: Harder, L.D., Barrett, S.C.H. (Eds.), Ecology and Evolution of Flowers. Oxford University Press, Oxford (UK), pp. 61–80.

Heslop-Harrison, Y., & Shivanna, K. R. (1977). The receptive surface of the angiosperm stigma. Annals of Botany, 41(6), 1233-1258.

Kearns, C. A., & Inouye, D. W. (1993). Techniques for Pollination Biologists. University Press of Colorado.

Kemp, J. E., & Vallejo‐Marín, M. (2021). Pollen dispensing schedules in buzz‐pollinated plants: experimental comparison of species with contrasting floral morphologies. American Journal of Botany, 108(6), 993-1005.

Knight, T. M., Steets, J. A., & Ashman, T. L. (2006). A quantitative synthesis of pollen supplementation experiments highlights the contribution of resource reallocation to estimates of pollen limitation. American Journal of Botany, 93(2), 271-277.

Luo, Z., Zhang, D., & Renner, S. S. (2008). Why two kinds of stamens in buzz‐pollinated flowers? Experimental support for Darwin's division‐of‐labour hypothesis. Functional Ecology, 22(5), 794-800.

Minnaar, C., Anderson, B., de Jager, M. L., & Karron, J. D. (2019). Plant-pollinator interactions along the pathway to paternity. Annals of Botany, 123(2), 225-245.

Ne'eman, G., Jürgens, A., Newstrom‐Lloyd, L., Potts, S. G., & Dafni, A. (2010). A framework for comparing pollinator performance: effectiveness and efficiency. Biological Reviews, 85(3), 435-451.

Ollerton, J., Alarcón, R., Waser, N. M., Price, M. V., Watts, S., Cranmer, L., et al. (2009). A global test of the pollination syndrome hypothesis. Annals of Botany, 103(9), 1471-1480.

Ouvrard, P., Quinet, M., & Jacquemart, A. L. (2017). Breeding system and pollination biology of Belgian oilseed rape cultivars (Brassica napus). Crop Science, 57(3), 1455-1463.

Pacini, E., & Dolferus, R. (2019). Pollen developmental arrest: maintaining pollen fertility in a world with a changing climate. Frontiers in Plant Science, 10, 679.

Pritchard, D. J., & Vallejo-Marín, M. (2020). Buzz pollination. Current Biology, 30(15), R858-R860.

Sampson, B. J., Stringer, S. J., & Marshall, D. A. (2013). Blueberry floral attributes and their effect on the pollination efficiency of an oligolectic bee, Osmia ribifloris Cockerell (Megachilidae: Apoidea). HortScience, 48(2), 136-142.

Thomson, J. D., Wilson, P., Valenzuela, M., & Malzone, M. (2000). Pollen presentation and pollination syndromes, with special reference to Penstemon. Plant Species Biology, 15(1), 11-29.

Vallejo‐Marín, M. (2019). Buzz pollination: studying bee vibrations on flowers. New Phytologist, 224(3), 1068-1074.

Vallejo-Marín, M. (2022). How and why do bees buzz? Implications for buzz pollination. Journal of Experimental Botany, 73(4), 1080-1092.

Vallejo-Marin, M., & Russell, A. L. (2023). Harvesting pollen with vibrations: Towards an integrative understanding of the proximate and ultimate reasons for buzz pollination. Annals of Botany, mcad189.

Vander Kloet, S. P. (1988). The Genus Vaccinium in North America. Agriculture Canada.

Webber, S. M., Garratt, M. P., Lukac, M., Bailey, A. P., Huxley, T., & Potts, S. G. (2020). Quantifying crop pollinator-dependence and pollination deficits: The effects of experimental scale on yield and quality assessments. Agriculture, Ecosystems & Environment, 304, 107106.

Wesselingh, R. A. (2007). Pollen limitation meets resource allocation: towards a comprehensive methodology. New Phytologist, 174(1), 26-37.

Experimental design

Materials and methods
Section 2.1
The cultivar Bluecrop should be introduced in the first paragraph of this section. Bluecrop is a cultivar of Vaccinium corymbosum, but the common name for the crop is “northern highbush blueberry”, thus this name should be given in this paragraph, along with the Latin name of the species in brackets (Vaccinium corymbosum; Ericaceae).

Section 2.2
The second paragraph 2.2 of this section “Observation of morphological characteristics of flower” should be completely removed and replaced with another paragraph 2.2 entitled “Categorization of flower life into four stages”. In this paragraph, the authors should describe how they grouped and organized their data (this is not a result, the paragraph 3.1 should be completely removed from the results): “The flower life was categorized into 4 stages: stage I, just before flower anthesis; stage II, flower just opened with the aperture not completely opened; stage III, flower with the aperture completely opened; stage IV, after corolla fall.”

Section 2.3
Replace the current title with “Dynamics of pollen release in relation to the flower age”. This paragraph should be completely written again. The information provided is useless, and even does not correspond to what the authors really did. Instead, the authors forgot to describe that they vibrated the flower with a toothbrush (or something like that, a tool commonly used to release pollen from flowers with poricidal anthers) before counting the pollen tetrads remaining in the anthers. Otherwise, I do not see how the anthers could lose pollen by themselves if they were bagged? In addition, the authors did not describe at all how they counted the number pf pollen tetrads remaining in the anthers. They authors should specify it by using the adequate reference (probably Dafni, 1992, 2005; or Kearns and Inouye, 1993) with the adequate standard protocol that they used for that.

Section 2.4
This section can be removed, as it is a standard description of SEM protocol. The figures can be kept for illustration in an appendix of the manuscript.

Section 2.5
Add “in relation to flower age” at the end of the title of this section. For the two protocols described, please provide the adequate standard protocol to what they refer with their adequate corresponding reference (probably Dafni, 1992, 2005; or Kearns and Inouye, 1993). In addition, please specify in this paragraph the definitions of each level of stigma receptivity intensity (+++, ++ etc.): e.g., how many bubbles, color of the stigma…

Section 2.6
Rewrite the title of the section as follows: “Foraging behavior and pollination behavior of insect foragers”. Lines 160-168, I did not understand this paragraph. For instance, I wonder if “style” should be replaced with “anther”, otherwise I do not understand. In addition, I suggest to rewrite this paragraph to make it clearer and more concise. As well for the next paragraph, lines 169-175, if the authors can rewrite it to make it clearer, while being concise and using the adequate scientific words (for instance, I am not sure that “crushed” and “saffron” are adequate, replace “dyed” with “stained”, probably “safranin” is better than “saffron”). I am not sure that specifying M and M’ values are required to explain what the authors did.

Section 2.7
Rewrite the title of the section as follows: “Dynamics of nectar secretion and flower visitation frequency by insect foragers in relation to flower age”. I have some little doubt that the flowers were really monitored continuously between 7:00 and 19:00, by who? By people? In that case the experiment would have required a lot of people during a very lot of time. If the authors have instead used cameras, please describe the monitoring with cameras. Line 182, an anther cannot be “pollinated”, and “immediately after the style” is meaningless, please rewrite the sentence. I do not understand all the end of the sentence “when the anthers were separated… until the end of the flowering period”, please rewrite it. “Nectar volume was measured every 24h with a micropipette”: in bagged flowers? In flowers left in open pollination so that bees could remove the nectar? Please specify. If it is in open pollination, the measurement is called “nectar standing crop” (see Corbet, 2003). If the flowers were bagged, the authors measured what is called the “nectar apparent secretion rate” (see Corbet, 2003).

Section 2.8
Please, rewrite the title of the section as follows: “Pollen limitation of fruit set”. In this experiment, the authors did not measure what they thought, which is basically called the “pollination requirements”, i.e., if cross-pollination increases the fruit set.
For the treatment (1), replace “natural pollination” with “open pollination”: ‘open pollination’ is the standard semantics in pollination ecology for this type of treatment.
For the treatment (2), replace “selfing” with “autonomous self-pollination”: again, ‘autonomous self-pollination’ is the standard semantics in pollination biology for this type of treatment. It corresponds to the spontaneous deposition of self-pollen onto the stigma. For blueberry, it corresponds actually mostly to parthenocarpy, as some cultivars can develop some seedless fruits through parthenocarpy (see DeVetter et al., 2022).

Treatments (3) and (4) are actually the same treatment: hand pollination with self-pollen. Self-pollen can come from the same flower, from flowers of the same plant, or from plants of the same cultivar (see DeVetter et al., 2022). This treatment can be compared with treatment (1) to assess pollen limitation, i.e., if the flowers were pollen limited to set a fruit. For the concept of (quantitative) pollen limitation, see Ashman et al. (2004), Knight et al. (2006), Wesselingh (2007), Harder and Aizen (2010), Webber et al. (2020). In addition, the experimental design was not adequate for assessing pollen limitation in blueberry: in that case, it is recommended to use hand pollinations with cross-pollen, so by using pollen from another cultivar (Aizen and Harder, 2007), by recording variables such as seed set or fruit mass, which are more accurate than fruit set to assess pollen limitation (Knight et al., 2006), by choosing the adequate experimental scale to avoid maternal resource reallocation (Knight et al., 2006; Wesselingh, 2007; Webber et al., 2020), and by avoiding maternal resource limitation (Knight et al., 2006; Wesselingh, 2007; Harder and Aizen, 2010) for instance by removing all the unused flowers. So perhaps this part can be removed from the manuscript.

Section 2.9
The authors should describe more what they analyzed, and with what analyses. In particular, we cannot see any statistical analysis in the results, figures, tables… (no statistics t, z or χ2, no P-value…) The authors can look at published papers in international journals to have a better of idea on how presenting and running statistical analyses.


References
Ashman, T. L., Knight, T. M., Steets, J. A., Amarasekare, P., Burd, M., Campbell, D. R., et al. (2004). Pollen limitation of plant reproduction: ecological and evolutionary causes and consequences. Ecology, 85(9), 2408-2421.

Corbet, S. A. (2003). Nectar sugar content: estimating standing crop and secretion rate in the field. Apidologie, 34(1), 1-10.

Dafni, A. (1992). Pollination Ecology: A Practical Approach. Oxford University Press.

Dafni, A., Kevan, P. G., & Husband, B. C. (2005). Practical Pollination Biology. Enviroquest.

DeVetter, L. W., Chabert, S., Milbrath, M. O., Mallinger, R. E., Walters, J., Isaacs, R., et al. (2022). Toward evidence-based decision support systems to optimize pollination and yields in highbush blueberry. Frontiers in Sustainable Food Systems, 6, 1006201.

Harder, L. D., & Aizen, M. A. (2010). Floral adaptation and diversification under pollen limitation. Philosophical Transactions of the Royal Society B: Biological Sciences, 365(1539), 529-543.

Knight, T. M., Steets, J. A., & Ashman, T. L. (2006). A quantitative synthesis of pollen supplementation experiments highlights the contribution of resource reallocation to estimates of pollen limitation. American Journal of Botany, 93(2), 271-277.

Webber, S. M., Garratt, M. P., Lukac, M., Bailey, A. P., Huxley, T., & Potts, S. G. (2020). Quantifying crop pollinator-dependence and pollination deficits: The effects of experimental scale on yield and quality assessments. Agriculture, Ecosystems & Environment, 304, 107106.

Wesselingh, R. A. (2007). Pollen limitation meets resource allocation: towards a comprehensive methodology. New Phytologist, 174(1), 26-37.

Validity of the findings

Results
Section 3.1
The authors should completely remove this part as it is completely useless: the morphology and basic flower biology of Vaccinium or blueberry was already described and with more information in Vander Kloet (1988), Courcelles et al. (2013), Sampson et al. (2013), the review DeVetter et al. (2022). And the last part of this section should be moved to the materials and methods (see the previous section in this review), as categorizing the flower life into four flower stages is not a result but a method used.

Section 3.2
This section should be entitled “Dynamics of pollen release in relation to the flower age”. Lines 220-223, remove the sentence “The results of the anther dehiscence…gradual pollen presentation”: it is useless, we already know that Vaccinium and blueberry have poricidal anthers (e.g., see Vander Kloet, 1988; DeVetter et al., 2022). The authors should focus on describing the gradual pollen release in relation with the flower age. The authors should also remove the stage IV from this result and from the Figure 4: the corolla fell with the anthers, so the authors could not count how many pollen tetrads remained in anthers that disappeared…

Section 3.3
The authors can completely remove this part. The pollen tetrad (and not pollen grain…) and stigma are not described precisely enough compared to the standards (e.g., see Brewbaker, 1967; Heslop-Harrison Shivanna, 1977; Pacini and Dolferus, 2019), and most of what the authors describe is already described (e.g., Vander Kloet, 1988; DeVetter et al., 2022).

Section 3.4
Replace “test results” with “in relation to flower age” in the title of this section. This paragraph should be completely written again, to be more concise, for instance as follows: “Pollen viability was of 52.8% (± SE?) at flower stage I, peaked at flower stage II at 79.2% (± SE?), and decreased at flower stage III at 38.2% (± SE?). Reaction with the benzidine hydrogen peroxide was high at flower stage I, higher at flower stage II with more bubbles etc.” The last sentence “Therefore, propagation…” is an interpretation, so it should appear in the discussion, not in the results.

Section 3.5
This section should be titled “Foraging behavior and pollination behavior of insect foragers”. The two first sentence should be removed, they are useless, they are not results, we already know it, and as long as it is not shown yet, Apis mellifera and Bombus spp. are both considered as ‘insect foragers’, not ‘pollinators’. It appears later in the paragraph that those two insect taxa deposit pollen on stigmas… Please, write again this section by being more concise and with better English formulations. The authors can remove ‘ligustica’ from the name Apis mellifera, Apis mellifera is enough, and to really show that the honey bees were ‘ligustica’, the authors should have confirmed it with genetic analyses… At last, the authors cannot use the expression ‘pollination efficiency’ for what they are referring to (see Ne’eman et al., 2010).

Section 3.6
Rewrite the title of the section as follows: “Dynamics of nectar secretion and flower visitation frequency by insect foragers in relation to flower age”. Line 258, replace “flower stage” with “flower life”, and “anther dispersal” with “pollen dispersal”.

Section 3.7
I think that the authors can completely remove this part: see my comments in the previous section ‘materials and methods’.

Discussion
Please, revise carefully the discussion accordingly to all the changes that have to be made to the manuscript following my previous comments.

Figures and Tables
Figure 1A can be kept in an appendix of the manuscript, not in the core manuscript as it is not important to understand the paper. Figures B, C and D can be removed: we already know how flowers and fruits of blueberry look like…

Figure 2 can be removed from the manuscript. As well, we already know how a blueberry flower looks like, and the image quality is not especially outstanding.

Figure 3: for the legend, please rewrite following my suggestions of formulation in the materials and methods section.

Figure 4: “pollen tetrads” instead of “pollen grains”. Remove the last bar “flower stage IV” (see my previous comment in the result section). For the legend, as well revise accordingly to my previous comments. The sentence “The number of pollen remaining …” is a result, so it should appear only in the text of the result section, not in a figure legend, so please remove it.

Figure 5: this figure can be kept in an appendix for illustration. But it is not a result (see my previous comments).

Figure 6: “pollen viability” instead of “pollen vitality”. As for the figure 4, please remove the second sentence which is a result, not a figure legend. And please remove the flower stage IV from the figure: the anthers fell with the corolla, so if the anthers disappeared, how is it possible to quantify pollen viability?

Figure 7: can be kept in an appendix of the manuscript, but the authors should not keep it in the core manuscript as these images are not required to understand the paper. Remove ‘ligustica’, and add ‘spp.’ after Bombus (as well as in the text of the manuscript everywhere).

Figure 8, legend: as well, review accordingly to my previous comments, and remove the second sentence which is a result, not a legend.

Figure 9, legend: same comments. Replace “Flowering duration” with “flower age”. Replace “Visiting frequency (times/flower/h)” with “Flower visit frequency (visits.flower-1.hour-1)”.

Table 1, legend: same comments, the second sentence is a result, not a legend. The row “number of treatments” should be removed from the table: this information should be added directly in the legend as follows: “Ten flowers were sampled for each flower stage”.

Table 2: should be removed (see my comments in the previous sections).


References

Brewbaker, J. L. (1967). The distribution and phylogenetic significance of binucleate and trinucleate pollen grains in the angiosperms. American Journal of Botany, 54(9), 1069-1083.

Courcelles, D. M. M., Button, L., & Elle, E. (2013). Bee visit rates vary with floral morphology among highbush blueberry cultivars (Vaccinium corymbosum L.). Journal of Applied Entomology, 137(9), 693-701.

DeVetter, L. W., Chabert, S., Milbrath, M. O., Mallinger, R. E., Walters, J., Isaacs, R., et al. (2022). Toward evidence-based decision support systems to optimize pollination and yields in highbush blueberry. Frontiers in Sustainable Food Systems, 6, 1006201.

Heslop-Harrison, Y., & Shivanna, K. R. (1977). The receptive surface of the angiosperm stigma. Annals of Botany, 41(6), 1233-1258.

Ne'eman, G., Jürgens, A., Newstrom‐Lloyd, L., Potts, S. G., & Dafni, A. (2010). A framework for comparing pollinator performance: effectiveness and efficiency. Biological Reviews, 85(3), 435-451.

Pacini, E., & Dolferus, R. (2019). Pollen developmental arrest: maintaining pollen fertility in a world with a changing climate. Frontiers in Plant Science, 10, 679.

Sampson, B. J., Stringer, S. J., & Marshall, D. A. (2013). Blueberry floral attributes and their effect on the pollination efficiency of an oligolectic bee, Osmia ribifloris Cockerell (Megachilidae: Apoidea). HortScience, 48(2), 136-142.

Vander Kloet, S. P. (1988). The Genus Vaccinium in North America. Agriculture Canada.

Annotated reviews are not available for download in order to protect the identity of reviewers who chose to remain anonymous.

---

## Round 0.2 · accepted · Accept

The manuscript is improved and can be accepted in its current state.

·

Basic reporting

The authors corrected the English grammar mistakes and made a fluent manuscript. ıt is now more understandable. They embedded imortant literatures into the MS, and clarified the methods section. Thanks a lot to the authors.

Experimental design

Aims are now more explanatory.
Methods became replicabe.

Validity of the findings

Results are better handled and used obligately

Additional comments

The manuscript is now sounds better and show important and useful results.
I want to thank to the authors for their neat and patient responses.
Now in my opinion the MS can be accepted. I made a few proposals to the revised MS and added it in the attachment.